# A Proximal Operator for Inducing $2{:}4$-Sparsity

## Abstract

Recent hardware advancements in AI Accelerators and GPUs allow to efficiently compute sparse matrix multiplications, especially when 2 out of 4 consecutive weights are set to zero. However, this so-called 2:4 sparsity usually comes at a decreased accuracy of the model. We derive a regularizer that exploits the local correlation of features to find better sparsity masks in trained models. We minimize the regularizer jointly with a local squared loss by deriving the proximal operator for which we show that it has an efficient solution in the 2:4-sparse case. After optimizing the mask, we introduce masked-gradient updates to further minimize the local squared loss. We illustrate our method on toy problems and apply it to pruning entire large language models up to 70B parameters. On models up to 13B we improve over previous state of the art algorithms, whilst on 70B models we match their performance.

## 1. INTRODUCTION

The extensive adoption of large language models has sparked renewed interest in post-training model compression to reduce inference cost and latency (Chitty-Venkata et al., 2023; Park et al., 2024). The most notable techniques are quantization of model weights and activations (Frantar et al., 2023; Dettmers et al., 2022; Frantar et al., 2024) as well as model pruning, i.e., the removal of weights or structures (Frantar & Alistarh, 2023; Sun et al., 2024; Ashkboos et al., 2024). However, pruning entire network structures like columns and rows of weight matrices leads to nonnegligible accuracy drops. Hence, the surge for more flexible sparsity patterns that allow to prune individual weights has sparked. This is accompanied by hardware and software support for efficient sparse matrix multiplications (Pool et al., 2021). In this work we consider structured sparsity, that

is we attempt to prune weights of linear models (or linear layers in a neural network) into a structure where out of each consecutive $M \in \mathbb{N}$ weights $N \in \mathbb{N}$ weights are set to zero. Modern GPUs and AI Accelerators can efficiently represent structured sparsity patterns to compress the memory footprint of the model and accelerate the matrix multiplications.

Recent work is generally designed for pruning to unstructured sparsity and has then been applied to structured patterns. In this work, instead, we design a family of regularizers that directly induce structured sparsity. The resulting regularized pruning problems are then solved using the celebrated proximal gradient method. Unlike existing pruning methods, our method induces sparsity *gradually* over the iterations, which can lead to better masks. The proximal gradient method requires solving a *proximal operator* in each iteration, which itself is a nonconvex problem.

The key theoretical contribution of this paper is to show that this non-convex *proximal operator* can be efficiently solved by solving three convex subproblems. We empirically show that these regularizers tend to identify more efficient sparsity masks, reduce the squared loss and can lead to better performance when applied to pruning entire large language models. Our approach natively applies gradient descent on the layer level while finding the mask and after freezing. Our key empirical contribution is that we apply such local gradient descent after masking to previous state-of-the-art methods (WandA and SparseGPT) and find that we can improve those out of the box. Since Wanda and SparseGPT methods are extensively adopted, we expect that the masked gradient updates will have significant impact.

We defer proofs to Appendix B.

## 2. PROBLEM AND RELATED WORK

Large Language Models (LLMs) based on the transformer architecture (Vaswani et al., 2017) have become the workhorse of language modelling. Recent architectures like Llama (Touvron et al., 2023) use decoder only models. Innovations on the attention calculation (Dao, 2023) or its design as grouped-query attention (Ainslie et al., 2023) have in few years already led to notable efficiency improvements. This moves the focus more on the classic matrix multiplication in linear layers. The idea here is to find an

---

[1]Anonymous Institution, Anonymous City, Anonymous Region, Anonymous Country. Correspondence to: Anonymous Author <anon.email@domain.com>. *Work initiated while being an intern at Amazon.

Preliminary work. Under review by the International Conference on Machine Learning (ICML). Do not distribute.

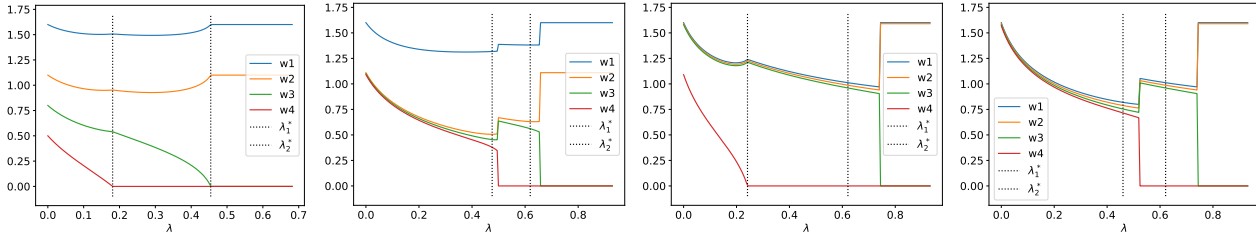

*Figure 1.* Illustration of the *regularization path*, e.g., optimal solution $(w_1, w_2, w_3, w_4)$ as a function of $\lambda$ in (7). From left to right, we are showing the result with (a) easy input $z = [1.6, 1.1, 0.8, 0.5]$ (b) nearly-tied $z_{2:4}$ with $z = [1.6, 1.11, 1.1, 1.09]$ (c) nearly-tied $z_{1:3}$ with $z = [1.6, 1.59, 1.58, 1.09]$ (d) All nearly tied with $z = [1.6, 1.59, 1.58, 1.57]$. The two dashed lines in all figures indicates the two critical thresholds $\lambda_1^*$ and $\lambda_2^*$ when the 3-sparse solution (and 2-sparse solution) become a *critical point* of (7), thus $\lambda \geq \lambda_1^*$ (or $\lambda \geq \lambda_2^*$) are necessary conditions for the solution to be 3-sparse (or 2-sparse), see Lemma 10 in Appendix B. Observe that in the "easy input" case, as we increase $\lambda$, the sparsification is *gradual*, rather than abrupt. In the "hard input" cases, decisions about those that are nearly tied are not made prematurely at smaller $\lambda$, the transition to exact structured sparsity happened at a much larger $\lambda$.

efficient approximation of the weight matrix such that the multiplication can be executed more efficiently as well as that the matrix can be stored efficiently in order to reduce the memory footprint of the matrix multiplication.

## 2.1. Matrix Compression

Traditional approaches involve low-rank approximations and sparsity. Modern hardware accelerators, which are used to run LLMs, cannot make use of sparsity if the 0s do not follow a regular structure. On the other hand, if a lot of structure is imposed, for example entire rows or columns are zeroed out, the accuracy drop is often too large.

A method aiming to strike a balance is so-called structured sparsity or $N{:}M$ sparsity. Here, out of $M$ consecutive weights $N$ are set to zero (Pool et al., 2021). For the case of 2:4 sparsity, one stores only two weights as well as a two-bit index per weight. Thus, for example, if the weights are kept in bfloat16 the memory footprint reduces from $4 \cdot 16 = 64$ bits to $2 \cdot (16 + 2) = 36$ bits, i.e., a compression to 56%. Also since those patterns are efficiently supported in hardware, it can reduce the FLOPs in matrix multiplications by a factor of up to 2x, which is mostly relevant during prompt encoding (aka *prefill*). LLM inference during decoding is on the other hand memory-bound rather than compute bound (Park et al., 2024). There the speedup comes from the memory compression. We note here that this speedup reduces when sparsity is combined with quantization (Jacob et al., 2018; Dettmers et al., 2022; Frantar et al., 2023), as the relative overhead of the position indices grows. With 2:4 sparsity the memory compression and decoding speedup for 16/8/4 bit datatype is 1.77x/1.6x/1.33x.

## 2.2. Pruning

Many recent works propose new algorithms and heuristics to prune the linear layers in LLMs. Whilst some focus on

sparsity (Frantar & Alistarh, 2023; Sun et al., 2024; Dong et al., 2024; Wei et al., 2023) others also remove entire structures, like heads, channels or layers resolving the need for special hardware support (Xia et al., 2024; Ma et al., 2023; Muralidharan et al., 2024). All of this methods have a non-negligible drop in performance metrics, requiring a delicate trade-off between latency/cost and performance degradation of the models. It is known that fine tuning after compression recovers some of the performance drop (Sun et al., 2024; Frantar & Alistarh, 2023; Dong et al., 2024) and recent work have also studied more extensive distillation with a small percentage of pretraining tokens, for example Minitron (Muralidharan et al., 2024) or Llama-3.2.[1]

## 2.3. One-Shot Pruning with Squared Loss

In this work we focus on a local one-shot setting (Frantar & Alistarh, 2023). We consider linear layers in a deep neural network. Let $N, M \in \mathbb{N}$, $N < M$ and $d_i, d_o \in \mathbb{N}$ be the feature input and output dimension, where we assume that $d_i$ is divisible by $M$. We call a weight matrix $W \in \mathbb{R}^{d_o \times d_i}$ to be $N{:}M$ sparse if each cell of $M$ consecutive weights contains maximally $N$ non-zero entries $\sum_{i=1}^{M} |W_{r,(k-1) \cdot M+i}|_0 \leq N$ for all $k \in [d/M]$ and for each row $r \in [d_o]$. We will denote the set of $N{:}M$ sparse matrices as $\mathbb{S}_{N:M}$, leaving the dependency on $d_i, d_o$ implicit.

We assume that the network has been trained to convergence resulting in dense weights $W^*$. Assuming that we have inputs to the liner layer $X \in \mathbb{R}^{d_i \times n}$ our goal is to find $N{:}M$ sparse weights that maintain the output of the layer as good as possible in terms of a squared loss (LeCun et al., 1989; Hubara et al., 2021; Frantar & Alistarh, 2023; Sun et al., 2024):

$$\underset{W \in \mathbb{S}_{N:M}}{\arg\min} \frac{1}{n} \|WX - W^*X\|_F^2. \tag{1}$$

---

[1]https://huggingface.co/meta-llama/Llama-3.2-1B#training-data

This problem can be solved for each row independently, because $N{:}M$ sparsity imposes the same level of sparsity across all rows and the Frobenius norm decomposes over the rows. Nonetheless, even for a single row, the problem becomes combinatorially hard. Whilst given a sparsity pattern, solving for the optimal weights is a convex problem, the number of masks that need to be searched is $\binom{M}{N}^{d_i/M}$. Since $N, M$ are determined by the hardware and fixed, the complexity grows exponentially with the matrix dimension and in practice we inevitably need to resort to heuristics.

We rewrite the loss in terms of the *Hessian* $H := \frac{XX^\top}{n}$ and using $\|A\|_F^2 = \mathrm{Tr}(AA^\top)$ as

$$\begin{aligned} L(W) :&= \frac{1}{n}\|WX - W^*X\|_F^2 \\ &= \mathrm{Tr}\left((W - W^*)H(W - W^*)^\top\right). \end{aligned} \quad (2)$$

We will also refer to this as *local squared loss* to emphasize that it is on a per-matrix level. Notice that on the diagonal of $H$ we simply have the mean squared activations of the corresponding input channels.

The $N{:}M$ sparse optimization problem simplifies significantly if the Hessian $H$ is diagonal, meaning that the input features are uncorrelated. In this case the weights cannot compensate for each other, that is each weight is either pruned to zero or kept at its original value. Given a pruning mask $M \in \{0,1\}^{d_o \times d_i}$ the loss is simply

$$\sum_{i,j}(1 - M_{i,j})W_{i,j}^* H_{j,j} W_{i,j}^* =: \sum_{i,j}(1 - M_{i,j})S_{i,j}^2, \quad (3)$$

and we can solve the minimization problem efficiently and optimally by simply pruning those weights with smallest scores $S_{i,j} := |W_{i,j}^* H_{j,j}^{1/2}|$. Although not derived in the way presented here, this is precisely the criterion that WandA (Weights and Activations) (Sun et al., 2024) proposes for pruning Large Language Models (LLMs) and use it as heuristic even when the Hessian is not diagonal.

**Theorem 1** (WandA is locally optimal for diagonal Hessians)**.** *Let the Hessian $H$ be diagonal, $W^*$ arbitrary and define $M^* \in \{0,1\}^{d_o \times d_i}$ such that for each $M$ cell it has zeros for the $N$ values with smallest score $S_{i,j}$. Then, $M^* \odot W^*$ is a minimizer of problem* (1)*.*

SparseGPT (Frantar & Alistarh, 2023) introduce an iterative heuristic that prunes the weight matrix in column blocks from left to right and takes off-diagonal elements of the Hessian into account. To determine which weights to prune, they use the criterion introduced by Hassibi & Stork (1992) and update the weights in the remaining right blocks, by using the inverse Hessian of the remaining sub-matrix. SparseGPT allows to efficiently prune multiple rows at once, without costly recomputations of the inverse Hessians.

Notice that both SparseGPT and WandA only require forward passes through the full model to populate the Hessian matrix or mean squared activations for each linear layer, which we dub *one-shot* setting (Frantar & Alistarh, 2023). Both algorithms run in a matter of minutes / hours on a single GPU on large language models up to scales of 100B parameters. We focus on aforementioned *one-shot* setting, where we only allow local updates, but no end-to-end tuning of the entire model. Further recent works one-shot pruning works include DSnoT Zhang et al. (2024), which provides small improvements in some cases over WandA and SparseGPT, and ALPS Meng et al. (2024), which is particularly suited for high sparsity.

## 2.4. Proximal Gradient

The proximal gradient (PG) method is a well-known optimization algorithm designed for solving composite minimization problems, where the objective function can be decomposed into two parts: a smooth differentiable function and a nonsmooth. Such problems arise frequently in many modern applications. In this setting, the objective function typically has the form:

$$\min_w\{f(w) + h(w)\},$$

where $f : \mathbb{R}^n \to \mathbb{R}$ is smooth and differentiable, while $h : \mathbb{R}^n \to (-\infty, \infty]$ is a possibly nonsmooth *structure-inducing* regularization term. To account for this nonsmooth component, the Proximal Gradient (PG) method incorporates a proximal operator alongside the standard gradient step for the smooth part, $f(w)$. The proximal operator, designed to mainly tackle nonsmooth functions, is defined as follows:

$$\mathrm{prox}_h(z) = \arg\min_w\left\{\frac{1}{2}\|w - z\|^2 + h(w)\right\}.$$

Therefore, to sum-up, starting with an arbitrary $w^0$, PG method with step-size $t > 0$ generates iteratively a sequence $\{w^k\}_{k \in \mathbb{N}}$ via the following iterative step:

$$w^{k+1} = \mathrm{prox}_{th}(w^k - t\nabla f(w^k)).$$

The PG method is particularly efficient when the smooth part $f$ has a Lipschitz-continuous gradient and when the proximal operator for $h$ has a closed-form solution (e.g., the hard-thresholding operator for $\ell_0$-regularized problems). However, if the proximal operator has no closed-form solution it might require an additional tailored optimization algorithm to solve the corresponding problem. Below, we will propose a new function to promote $N{:}M$ sparsity and discuss in detail its 2:4 proximal operator.

Proximal gradient methods have been extensively studied, particularly in the context of convex optimization problems. For a thorough overview of their theoretical guarantees and extensions, refer to (Beck, 2017) and references therein.

## 3. PROXIMAL OPERATOR FOR N:M SPARSITY

WandA (Sun et al., 2024) and SparseGPT (Frantar & Alistarh, 2023) were both designed primarily for unstructured sparsity and then simply applied to the $N{:}M$ sparse case by adding these additional constraints. Inspired by PG methods we propose regularizers that are explicitly designed for $N{:}M$ sparsity and iterate gradient updates on the squared loss (done on the matrix level) with the solution of the proximal operator (which decomposes into the $M$ cells). This leads to a gradual emergence of the pruning mask, where the emerging pattern of one cell can in fact influence the pruning pattern of other cells and avoids committing to a sparsity pattern to early. WandA and SparseGPT do not have this possibility. Our approach will consume more compute and time, but this is well invested. Compared against the cost of training the models and the inference cost, the additional cost to find a better mask and model is well invested.

### 3.1. Gradient Descent

Each step of PG requires a gradient step on the unregularized loss (2), which has a simple closed-form, see Appendix A:

$$W \leftarrow W - \eta \cdot 2(WH - W^*H), \qquad (4)$$

where we can use the largest eigenvalue of the Hessian to set the stepsize $\eta = \frac{1}{2\gamma_{\max}(H)}$, guaranteeing convergence. While each gradient step has complexity $O(d_i^2 d_o)$, it allows us to efficiently use the full parallelization of modern GPUs.

We will use gradient descent during PG, but we also propose it as a method to improve *any* local pruning method. Once the pruning mask $M$ is fixed, we can partially compensate for the pruning loss, by masked gradient updates:

$$W_M \leftarrow W_M - \eta \cdot 2M \odot (W_M H - W^*H). \qquad (5)$$

Our first contribution is to show that WandA and SparseGPT do benefit from such gradient updates after they completed the pruning.

### 3.2. Regularization and Proximal Operator

To induce $N{:}M$ sparsity, we define the following family of regularizers[2]:

$$r_{N:M}(w) := \sum_{\substack{\mathcal{S} \subset [M] \\ |\mathcal{S}| = N+1}} \prod_{j \in \mathcal{S}} |w_j|. \qquad (6)$$

**Fact 2** (The null space of the regularizer). *For all $w \in \mathbb{R}^M$ we have*

$$r_{N:M}(w) = 0 \text{ if and only if } \|w\|_0 \leq N.$$

[2]In Appendix C we propose a few more, less elaborate, regularizers for 2:4 sparsity.

The corresponding proximal operator is

$$\text{prox}_{\lambda r_{N:M}}(z) := \arg\min_{w \in \mathbb{R}^M} \left\{ \frac{1}{2}\|w - z\|^2 + \lambda r_{N:M}(w) \right\}.$$

The regularizer $r_{N:M}$ promotes $N{:}M$ sparsity in that $N$ sparse solutions are in the null space of this regularizer. Moreover, as $\lambda \to \infty$, there always exists $\lambda^*$ (as a function of input $z$) such that when $\lambda \geq \lambda^*$, the solution of the proximal operator becomes exactly $N$-sparse (see Figure 1 for an illustration).

For $2{:}4$ sparsity, the proximal operator of interest is

$$\arg\min_{w \in \mathbb{R}^4} \left\{ \frac{1}{2}\|w - z\|^2 + \lambda(|w_1||w_2||w_3| + |w_2||w_3||w_4| \right.$$
$$\left. + |w_3||w_4||w_1| + |w_4||w_1||w_2|) \right\}. \qquad (7)$$

The proximal operator can be used to induce 2:4 sparsity in all settings including: layerwise pruning, finetuning and pretraining using proximal gradient methods or straight-through gradient methods. We focus on layerwise pruning via the squared loss (2). The combined loss function for the 2:4 sparse case is then

$$L_\lambda(W) := L(W) + \lambda \sum_{w \in W} r_{2:4}(w), \qquad (8)$$

where $w \in W$ runs over all 2:4 cells of $W$. Following Fact 2 the matrix $W$ is 2:4 sparse if and only if the regularizer is zero. Importantly the proximal operator corresponding to the regularizer of multiple cells decomposes into multiple 2:4 proximal operators and hence the complexity of solving the proximal operator at each iteration of PG only grows linearly with the dimensionality of the matrix.

### 3.3. Solution of the 2:4-Proximal Operator

Solving the 2:4 proximal operator problem is tricky. It is non-convex, non-smooth and appears to require an exhaustive search-style approach. Nonetheless, we have identified interesting mathematical structure of this problem which led to a discovery of very efficient solutions to this problem with provable guarantees.

We first prove that we can always reduce the proximal operator to the case where $z$ is non-negative and sorted $z_1 \geq z_2 \geq z_3 \geq z_4 \geq 0$.

**Lemma 3.** *To solve problem (7), it suffices to solve*

$$\arg\min_{w \in \mathbb{R}_+^4} f(w), \quad with \qquad (9)$$

$$f(w) := \left\{ \frac{1}{2}\|w - z\|^2 + \lambda(w_1 w_2 w_3 + w_2 w_3 w_4 \right.$$
$$\left. + w_3 w_4 w_1 + w_4 w_1 w_2) \right\}, \qquad (10)$$

where $z_1 \geq z_2 \geq z_3 \geq z_4 \geq 0$. *Moreover, any optimal solution $w^*$ of problem* (9) *satisfies that $w_1^* \geq w_2^* \geq w_3^* \geq w_4^* \geq 0$.*

Observe that $f(w)$ is smoothly differentiable, but now we have a constrained optimization problem. The same result can generally be stated for $N$:$M$-sparsity.

Let us define the restricted version of problem (9) when $w_4$ is set to 0.

$$\arg\min_{w \in \mathbb{R}_+^3} g(w), \quad \text{with} \tag{11}$$

$$g(w) := \frac{1}{2}\sum_{i=1}^{3}(w_i - z_i)^2 + \lambda w_1 w_2 w_3. \tag{12}$$

As a side remark, the proximal operator for $r_{2:3}$ can be reduced to solving (11).

**Lemma 4** (Optimality conditions). *Assume $z_1 \geq z_2 \geq z_3 \geq z_4 > 0$. The solution $w^*$ is one of the following three cases.*

- *2-sparse: in which case $w^* = [z_1, z_2, 0, 0]$.*
- *3-sparse: in which case $w_4^* = 0$ and $w_{1:3}^*$ satisfies*

$$0 < w_1^* = z_1 - \lambda w_2^* w_3^*,$$
$$0 < w_2^* = z_2 - \lambda w_1^* w_3^*,$$
$$0 < w_3^* = z_3 - \lambda w_1^* w_2^*.$$

- *Dense: in which case $w^* = [w_1^*, w_2^*, w_3^*, w_4^*] > 0$ satisfies that*

$$w_i^* = z_i - \lambda \cdot \sum_{\{j_1, j_2\} \subset \{j \in [4], j \neq i\}} w_{j_1}^* w_{j_2}^*.$$

In Appendix B.4 we also include necessary conditions on $\lambda$ for the solutions to be 2-/3-sparse. Whilst the 2-sparse solution is trivial, optimizing for the 3-sparse and dense solution requires more care. We next turn to the Hessians of $f$ and $g$, see Equation (23) and Equation (24) for their explicit forms.

**Lemma 5** (Properties of the Hessians).

- *The set $\mathcal{C}_3 := \{w \in \mathbb{R}^3 \mid \nabla^2 g(w) \succeq 0\}$ is convex.*
- *The set $\mathcal{C}_4 := \{w \in \mathbb{R}^4 \mid \nabla^2 f(w) \succeq 0\}$ is convex.*

This property of the Hessian is crucial, as it allows us to focus on finding a single minimum.

**Corollary 6** (No spurious local minima.). *The set of all local minima of $f(w)$ is convex. Moreover, if there exists $w^* \in \arg\min_{w \in \mathbb{R}_+^4} f(w)$ such that $w_i^* > 0$ for all $i \in [4]$ (i.e., in the "Dense" case from Lemma 4), then all local minima of $f(w)$ are global minima.*

*The set of all local minima of $g(w)$ is convex. Moreover, if there exists $w^* \in \arg\min_{w \in \mathbb{R}_+^3} g(w)$ such that $w_i^* > 0$ for all $i \in [3]$, then all local minima of $g(w)$ are global minima.*

To solve the proximal operator we can thus solve the following constrained optimization problems

$$\min\left\{g(w) \mid w \in \mathbb{R}^3, w \geq 0, \nabla^2 g(w) \succeq 0\right\}, \tag{13}$$

$$\min\left\{f(w) \mid w \in \mathbb{R}^4, w \geq 0, \nabla^2 f(w) \succeq 0\right\}. \tag{14}$$

The optimal solutions to (13) and (14) are local minima to (11) and (9) respectively if the solutions have gradient $= 0$. In general, the set of local minima for a non-convex problem can be scattered all over the place, but with Corollary 6 we have shown that for these special functions $f$ and $g$, the local minima (if they exist at all) form a convex set and have the same objective values.

Problems (13) and (14) are convex optimization problems due to Lemma 5. They can be solved using interior point methods with self-concordant barrier functions. As barrier functions, we can use $-\log\det\nabla^2 f(w)$ and $-\log\det\nabla^2 g(w)$ respectively, as well as $-\log(w_i)$ to encode the positivity constraints (Boyd & Vandenberghe, 2004, Chapter 11.2).

We have now reduced our initial non-convex problem to three convex subproblems, out of which one has a trivial solution. We can then solve those three problems and select the minimizer among the three cases.[3]

**Theorem 7.** *Algorithm 1 returns an optimal solution to* (9).

---

**Algorithm 1** Solve Prox with decreasing non-negative input

**Require:** $z_1 \geq z_2 \geq z_3 \geq z_4 \geq 0$
1: **procedure** PROXENUMERATE($z, \lambda$)
2:     $w^{(2)} \leftarrow [z_1, z_2, 0, 0]$
3:     $w^{(3)} \leftarrow [\text{solution to } (13), 0]$.
4:     $w^{(4)} \leftarrow \text{solution to } (14)$.
5:     **return** $\arg\min_{w \in \{w^{(2)}, w^{(3)}, w^{(4)}\}} f(w)$
6: **end procedure**

---

### 3.4. Generalization to $N$:$M$ sparsity.

2:4 sparsity is currently the most relevant sparsity pattern. It is plausible that general $N$:$M$ sparsity patterns also benefit from a more gradual pruning algorithm. For 1:$M$ sparsity, the proximal operator becomes very simple as it is just a quadratic function and it can be solved in closed form. However, for more general patterns when $N > 2$ solving the proximal operator becomes more challenging. Lemma 4 still holds and it is enough to consider the non-negative sorted case. However, finding the minimizers is harder. In particular it is not obvious how one could replace Lemma 5,

---

[3]If $w^{(4)}$ in Section 3.5 is a local minimum of $f$, by Corollary 6 this suffices to know it is the global optimum and there is no need to compute $w^{(2)}, w^{(3)}$. For ease of notation and to parallelize when solving multiple cells, we nonetheless always compute the three cases.

i.e., the convexity of the space where the Hessian is convex. There we used that the Hessian in the 2:4 sparse case is linear in $w$, which will no longer hold for $N > 3$. There might be different arguments to show that those cases can be handled efficiently as well.

### 3.5. Parallelized Prox by Gradient Descent

We showed that Equations (13) and (14) are convex optimization problems that can be solved with guarantees for example with interior point methods with self-concordant barrier functions. However, in order to apply this algorithm at scale we need to optimize it further. For example when pruning the MLP down projection of a Llama 70B model (Dubey et al., 2024), the matrix has dimensions 28672 and 8192, meaning that it has around 59 million cells of four weights. Thus at each iteration of Proximal Gradient, we have to solve Algorithm 1 59 million times, which is infeasible if not done in a parallelized fashion using modern GPUs.

We will thus resort to a gradient descent solver of which we conjecture that it always correctly solves the problem. We will focus on the case of (14), and the case for (13) follows similarly. The constraints $w \geq 0$ is easily enforced by projected gradient descent. The second constraint $\nabla^2 f(w) \succeq 0$ is more computationally expensive and we do not want to compute it at every iteration. Luckily, empirically we find that we actually do not need to enforce this constraint at all. First note two facts

**Fact 8.** *a)* $w = [0, 0, 0, 0]$ *is always a feasible point of* (14). *b) Let* $\gamma_{max}$ *denote the largest eigenvalue of a matrix. For all* $w \in \mathbb{R}_+^4$ *we have* $\nabla^2 f(w) \succeq 0 \Rightarrow \gamma_{max} \left[ \nabla^2 f(w) \right] \leq 4$.

Thus, whilst we are in the convex region of the loss, the gradients are Lipschitz with constant 4. Using $1/4$ as step size, we are guaranteed to never cross a local minimum as long as we stay in the convex region. In our extensive evaluations we observe that indeed gradient descent does never exit the region of convexity *if the global minimum is dense*. This implies that we can run gradient descent without the need of a barrier function. Whilst we could not find any counterexamples we have not been able to formally prove this and hence state is explicitly as a conjecture.

**Conjecture 9.** *If the minimizer* $w^*$ *of* (14) *fulfills* $w_i^* > 0$, *then*

$$w^0 = [0, 0, 0, 0]; \quad w^{k+1} = \max[w^k - \eta^k \nabla f(w^k), 0],$$

*with* $\eta = 1/4$ *and the maximization done elementwise to enforce* $w^{k+1} \in \mathbb{R}_+^4$ *converges to* $w^*$. *Furthermore, any intermediate point* $w^k$ *is in the feasible set of* (14).

The same considerations and empirical insights apply to the 3-sparse case. Hence, in practice when computing $w^{(3)}$

---

**Algorithm 2** Matrix Prox Pruner 2:4

1: **procedure** PRUNEPROX($W^*, H, \{\lambda_k\}_k$)
2:     $W \leftarrow W^*, \eta \leftarrow \frac{1}{2\gamma_{max}(H)}, k \leftarrow 0$
3:     **while** $W \notin \mathcal{S}_{2:4}$ **do**
4:         $W \leftarrow W - \eta \cdot 2(HW - HW^*)$
5:         $W, s, p \leftarrow \texttt{PosSort}(W)$
6:         $W \leftarrow \texttt{ProxEnumerate}(W, \lambda_k)$
7:         $W \leftarrow \texttt{InvPosSort}(W, s, p)$
8:         $k \leftarrow k + 1$
9:     **end while**
10:     **return** $W$
11: **end procedure**

---

and $w^{(4)}$ in Algorithm 1 we run the gradient descent based optimization without enforcing to remain in the region of convexity. As soon as we witness that we moved out of the convex region, we can stop the optimization as by Conjecture 9 and Corollary 6 the respective case can not be the optimal one. In practice, we stop when witnessing that the gradient norm increases, as this cannot happen whilst being in the convex region with our choice of step size. Therefore, we slightly deviate from Algorithm 1 in that in line 3 and 4 we do not compute the exact minima if we witness its not the the minimum of the three cases anyways. If Conjecture 9 is true, then Theorem 7 holds even with this modification.

In our implementation, the GD-based Algorithm 1 is more than 10x faster than our implementation of the interior point method-based Algorithm 1, and they always obtain numerically the same solution.

### 3.6. Full Algorithm

We can now put together the full matrix pruning algorithm, see Section 3.5. We define two functions: $\texttt{PosSort}(W)$ that returns the weights such that each 2:4 is sorted and without sign as well as $s, p$ that indicate the sign and original position of the weights. $\texttt{InvPosSort}(W, s, p)$ which undoes above operation. Furthermore, we use a schedule $\{\lambda_k\}_k$ for the regularization such that $\lambda_k \to \infty$. In this work we consider exponential schedules $\lambda_k = \lambda_0 \beta^k$, for $\lambda_0 > 0$ and $\beta > 1$.

## 4. EXPERIMENTS

For our experiments unless otherwise stated we use $\lambda_k = \lambda_0 \beta^k$ with $\lambda_0 = 0.01$ and $\beta = 1.01$. After ending Proximal Gradient, we do 1000 steps of masked gradients according to (5) to minimize the local squared loss. Furthermore, we initialize all methods *WandA* style by transforming $W_{i,j}^* \mapsto W_{i,j}^* H_{j,j}^{1/2}$, $H_{i,j} \mapsto H_{i,j} H_{j,j}^{-1/2} H_{i,i}^{-1/2}$. This helps the proximal pruning to not commit to weights that seem important solely in terms of magnitude, but takes the re-

spective mean squared activations into account (Sun et al., 2019). After pruning this transformation is reversed. All experiments can run on a single NVIDIA A100 GPU with 40GB of memory, but we used multiple GPUs in parallel to speed up the experimentation.

### 4.1. Toy Experiments

We start with a simple experiment to illustrate the inner workings of our algorithm and the shortcoming of WandA and SparseGPT. We consider the 2:4 sparse case of a single row with dense weights $w^* = (0, 5, 3, 2,\ 0, 5, 5, 2)$. We further assume that the Hessian has ones on the diagonal and zero elsewhere except that the fourth and eighth weight are perfectly correlated (value 1 on the respective off-diagonal elements). In this case the optimal 2:4 sparse solution is $w = (0, 5, 0, 4,\ 0, 5, 5, 0)$. The difficulty here is that the fourth and eighth weight if looked at individually within their 2:4 cell, would be pruned, but instead we can merge them into one weight and prune another weight incurring a smaller loss.

Since WandA (Sun et al., 2024) completely ignores the correlations, it prunes to $w = (0, 5, 3, 0,\ 0, 5, 5, 0)$, which is suboptimal. SparseGPT (Frantar & Alistarh, 2023) would in a first step prune the first $'2'$ because it can completely absorb it into the last weight due to their correlation $w \mapsto w = (0, 5, 3, 0,\ 0, 5, 5, 4)$. However then in the second step, when pruning the second cell, it drops the combined $'4'$ because it incurs lower loss than pruning one of the $'5's$ $w \mapsto w = (0, 5, 3, 0,\ 0, 5, 5, 0)$. Therefore, the SparseGPT heuristic can lead to deferred loss.

Our proximal optimizer in turn adapts to the situation. In the second cell, the $'2'$ is quickly pushed to 0. This in turn increases the gradient magnitude of the first $'2'$, pushing it eventually over the value of the $'3'$ in the first cell. To simplify the understanding, we animated the evolution of the weights with a gif $\rightarrow$ LINK.

Next we consider the effect of correlation between input features more structurally. Therefore we generate synthetic Hessians and random weights:

```
d=1024
diag = torch.diag(torch.rand(d))
Z = torch.randn(d, d) / float(d)**(1/2)
Z = Z @ Z.T
hessian = alpha * diag + (1 - alpha) * Z
weights = torch.randn(1, d)
```

We then prune with all the considered methods and after masking optimize the weights with gradient descent. Results are discussed and shown in Figure 2.

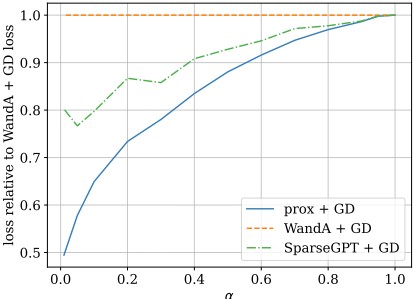

*Figure 2.* Toy experiment with generated weights and Hessians. Without correlation ($\alpha = 1$) all considered pruning methods result in the same mask and loss. As we increase the correlations, our proximal approach leads to the best mask as shown by the smallest loss.

### 4.2. Scheduling $\lambda$

During the course of proximal gradient, we increase the regularizer $\lambda$ with an exponential schedule to guarantee, that no matter what input, eventually the weights will be 2:4 sparse. In this work we consider exponential schedules $\lambda_k = \lambda_0 \beta^k$, for $\lambda_0 > 0$ and $\beta > 1$. Clearly, good choices of those hyperparameters depend on each problem and are expensive to ablate when done for each problem separately. We show the impact of different choices on the local loss and the runtime in the Appendix Figure 3. Notice that the optimization problem (9) is not scale invariant when scaling $z$ alone. However, if $w^*$ is the optimal solution to the problem $z, \lambda$, then $cw^*$ is the optimal solution for the problem $(cz, \lambda/c)$. Since it is too expensive to extensively search the hyperparameters for each problem individually, we propose to consider

$$\lambda_k = \frac{\tilde{\lambda}_0}{\text{Mean}[|W^*|]} \beta^k. \tag{15}$$

### 4.3. Pruning Large Language Models

To prune LLMs, we use 2 million tokens from the c4 dataset (Raffel et al., 2020), which we pack with end-of-sequence tokens and chunk into 1024 sequences of lengths 2048. We estimate all Hessians from the unpruned model, and do not propagate the pruning errors as we find that this has little impact (Appendix D.1). As baselines we run WandA (Sun et al., 2024) and SparseGPT (Frantar & Alistarh, 2023) both with their original algorithms as well as with our innovative additional 1000 gradient descent steps after masking, see Section 3.1, where we observed that the local loss has usually converged after 1000 steps as shown in Appendix Figure 5. We also run DSnoT (Wanda) Zhang et al. (2024). For the proximal, for small models we used our default setting $\lambda_k = \lambda_0 \beta^k$ with $\lambda_0 = 0.01$ and $\beta = 1.01$. For the 70B models, we observed that this would require more than 2000 iterations of PQ. We thus used the heuristic from Equa-

*Table 1.* Validation Perplexity of OpenLlama (3B/7B/13B) and Llama3.1 (8B/70B) (Pretrained/Instruct). . All the methods with +*GD* are an innovation of the present work.

| Method | 3b_v2 C4 | Wiki | 7b_v2 C4 | Wiki | 13b C4 | Wiki | 8B C4 | Wiki | 8B Inst C4 | Wiki | 70B C4 | Wiki | 70B Inst C4 | Wiki |
|---|---|---|---|---|---|---|---|---|---|---|---|---|---|---|
| dense | 9.68 | 14.15 | 8.84 | 12.15 | 8.11 | 11.52 | 9.68 | 7.93 | 11.28 | 7.27 | 7.32 | 3.34 | 8.26 | 3.71 |
| wanda | 29.48 | 60.32 | 17.63 | 27.02 | 13.17 | 35.45 | 36.32 | 27.26 | 43.30 | 26.67 | 13.42 | 10.25 | 13.68 | 8.74 |
| *wanda+GD* | 18.23 | 33.05 | 14.35 | 21.85 | 11.73 | 18.17 | 22.19 | 20.46 | 28.45 | 20.71 | 12.32 | 10.38 | 13.08 | 8.91 |
| sp.gpt | 18.76 | 33.78 | 14.24 | 22.29 | 11.91 | 19.51 | 24.95 | 23.18 | 35.55 | 26.19 | 13.00 | 10.93 | 14.05 | 9.37 |
| *sp.gpt+GD* | 16.72 | 29.58 | 13.31 | 20.83 | 11.43 | 17.19 | 21.00 | 19.86 | 27.54 | 20.85 | 12.06 | 10.49 | 13.06 | 8.91 |
| DSnoT | 26.54 | 44.17 | 17.62 | 26.17 | 13.71 | 32.89 | 40.66 | 30.40 | 48.80 | 31.46 | 14.34 | 10.54 | 14.51 | 9.16 |
| *prox+GD* | 16.27 | 28.59 | 13.19 | 20.55 | 11.40 | 17.71 | 20.86 | 19.83 | 26.46 | 20.20 | 12.33 | 10.38 | 13.12 | 8.91 |

tion (15) and the results of Figure 3 and selected $\beta = 1.005$, $\tilde{\lambda}_0 = 1 \cdot 10^{-3}$ to strike a good balance between performance and runtime.

We prune models from the openLlama family (Geng & Liu, 2023) and the Llama-3.1 models (Dubey et al., 2024), see Table 1. We evaluate the models both on in-distribution validation data from c4, as well as out of distribution data from WikiText2 (Merity et al., 2016).

On models up to 13B parameters, we see that the proximal approach consistently outperforms the other methods at least in-distribution (C4) and mostly also on wikitext data (wiki). In the appendix Figure 4 we show that this is related to an improvement in local squared loss. Furthermore, WandA and SparseGPT both benefit clearly from our proposed masked gradient descent based updates after pruning with the respective strategy.

Interestingly, whilst prox tends to give the best perplexity, it turns out that our innovation on using masked gradient updates (Section 3.1 is empirically more relevant, as it consistently improves the perplexity on C4. As a by-product of our work we thus identified a method that can be plugged on top of existing SOTA one-shot pruning methods. In Table 3 we evaluate the pruned Llama-3.1 8B Instruct model on 6 Downstream tasks and find that on average we improve the SOTA more than 3%.

On the 70B models, whilst the gradient updates continue to improve the in-distribution loss, it can harm the wiki perplexity in case of Wanda. Furthermore on the 70B models we see that the method to find the mask has a very minor impact on the final perplexity.

Overall, we find that a 2:4 sparse model is consistently worse than the dense smaller model of the same class which is consistent with the results on Llama 1 and 2 models found by Sun et al. (2024, Table 3)

## 5. Discussion

We have introduced a method to gradually induce structured sparsity in pretrained linear layers, for example of large language models. The immediate objective is to minimize the linear squared loss at the matrix level. We showed that a local improvement of the squared loss leads to an improvement in model perplexity. On the theoretical side, we showed how to efficiently solve the complex proximal operator for 2:4 sparsity.

On the empirical side, we observe that the sparsity mask does not make a huge difference for recent LLMs, but that our introduced local gradient based updates can also be used on top of existing local pruning methods and improve their performance. In particular, our proposed masked-gradient updates can easily be used to improve the widely adopted methods Wanda and SparseGPT. We believe that the masked gradient updates will be widely adopted.

Whilst we thus improved the state of the art on one-shot pruning, the resulting pruned models clearly are not good enough to be of direct practical relevance. Looking at Table 1, we see that the pruned 13B/7B models fall clearly behind the dense 7B/3B model. However, for a 2x matrix compression this should be the lowest bar a model has to take in order to be useful. This is even more because whilst the structured sparsity reduces the latency and footprint of matrix multiplications, it does not decrease the sizes of in-/output activations or the KV-cache.

One benefit of the proximal approach and the theoretical insight we provided is that they can also be applied during pretraining together with a regular pretraining objective.[4] Other pruning approaches discretely interrupt the optimization, whilst the proximal operator is applied at every step, hence gradually pushing the model to sparsity. This could pave the road for more powerful sparse models with less quality degradation.

---

[4]Concurrent work Fang et al. (2024) found that end-to-end learning of masks is beneficial also post training.

## Impact Statement

This paper presents work whose goal is to advance the field of Machine Learning. There are many potential societal consequences of our work, none which we feel must be specifically highlighted here.

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

## A. Gradient and Hessian

We have the following loss function Equation (2):

$$L(W) = \mathrm{Tr}\left((W - W^*)H(W - W^*)^\top\right).$$

The loss is simply the sum of the loss of each individual row and we e can minimize this on a per-row basis ($w \in \mathbb{R}^{d_i}$)

$$L(w) = (w - w^*)^\top H(w - w^*). \tag{16}$$

From here we can simply compute the gradients:

$$\nabla L(w) = 2H(w - w^*) = 2Hw - 2Hw^*. \tag{17}$$

We can the stack this again for the whole matrix and obtain:

$$\nabla L(W) = 2WH - 2W^*H. \tag{18}$$

Notice that the second term can be precomputed once, and does not need to be computed again at each iteration. Computing the second order derivatives per row we find $\nabla^2 L(w) = 2H$, hence also our naming conventions (ignoring the factor of 2).

## B. Proofs

### B.1. Proof of Theorem 1

*Proof.* We first show that a diagonal Hessian implies that for any mask, the optimal values of the remaining weights match their dense counterparts. Therefore, let $M$ be an arbitrary binary mask and assume that the Hessian is diagonal, i.e., $H_{ij} = 0$ whenever $i \neq j$. We use the formulation of Equation (2) where we explicitly add the mask

$$L(W, M) = \mathrm{Tr}\left((M \odot W - W^*)H(M \odot W - W^*)^\top\right) \tag{19}$$

$$= \sum_{i,j}(M_{i,j}W_{i,j} - W^*_{i,j})^2 H_{j,j} \tag{20}$$

$$= \sum_{i,j|M_{i,j}=1}(W_{i,j} - W^*_{i,j})^2 H_{j,j} + \sum_{i,j|M_{i,j}=0}(W^*_{i,j})^2 H_{j,j}. \tag{21}$$

Notice that the second term solely depends on the mask and not on the weights of $W$. Furthermore, the first term can be set to zero by matching the original weights. By definition WandA selects a mask that minimizes the second term whilst keeping the other weights fixed, hence minimizing the overall squared loss. □

### B.2. Proof of Fact 2

*Proof.* The "if" part is trivial. If $\|w\|_0 \leq N$, then for any index subset $\mathcal{S} \subset [M]$ with $|\mathcal{S}| = N + 1$, at least one of the coordinate of $w_\mathcal{S}$ is 0. To see the "only if" part, if suffices to prove the contra-positive. If $\|w\|_0 > N$, let $\tilde{\mathcal{S}}$ to be the first $N + 1$ coordinates of $w$ in $\tilde{\mathcal{S}}$, thus $r_{N:M}(w) \geq \prod_{i \in \tilde{\mathcal{S}}}|w_j| > 0$. □

### B.3. Proof of Lemma 3

*Proof.* First observe that the objective function of problem (7) is invariant to joint permutations and signs of $w$ and $z$. In other words, if $\tilde{w}^*$ is an optimal solution of problem (7) for a given input $\tilde{z} \in \mathbb{R}^4$, then for any permutation $\Pi$ and any diagonal matrix $D$ with diagonal elements being $-1$ or $1$, the $D\Pi\tilde{w}^*$ is an optimal solution of problem (7) when the input is $D\Pi\tilde{z}$ and having the same optimal value. Therefore, without the loss of generalization, we can choose $\Pi$ to be the permutation that sorts the input into a descending order by the absolute value of $|z|$, and $D = \mathrm{diag}(\mathrm{sign}(\Pi z))$.

Next, we observe that for a non-negative input $z$, any optimal solution of problem (7) is always non-negative. To see this, if $w_i^*$ is negative and $z_i > 0$, replacing it with $|w_i^*|$ retains the regularizer but $(|w_i^*| - z_i)^2 < (w_i^* - z_i)^2$. If $z_i = 0$, the corresponding optimal solution is $w_i^* = 0$, which is non-negative. Thus, we can remove the absolute values and focus on solving problem (9).

Moreover, we observe that a sorted (descent order) input $z$ implies that any optimal solution is also sorted in a descent order. Indeed, we take $w^*$ to be an optimal solution with $w_i^* < w_j^*$ for some $1 \leq i < j \leq 4$. Since the regularizer is invariant to permutations, we focus on the quadratic part and show that in this case $\tilde{w}$, which is defined by $\tilde{w}_i = w_j^*$, $\tilde{w}_j = w_i^*$ and $\tilde{w}_k = w_k^*$ for all $k \neq i, j$, we have that

$$
\begin{aligned}
\|w^* - z\|^2 - \|\tilde{w} - z\|^2 &= (w_i^* - z_i)^2 + (w_j^* - z_j)^2 - (\tilde{w}_i - z_i)^2 - (\tilde{w}_j - z_j)^2 \\
&= (w_i^* - z_i)^2 + (w_j^* - z_j)^2 - (w_j^* - z_i)^2 - (w_i^* - z_j)^2 \\
&= 2(w_j^* z_i + w_i^* z_j - w_i^* z_i - w_j^* z_j) \\
&= 2(w_j^* - w_i^*)(z_i - z_j) \\
&\geq 0,
\end{aligned}
$$

where the last inequality follows from the facts that $w_i^* < w_j^*$ and $z_i \geq z_j$. This shows that $\tilde{w}$, which is a sorted (descent order) vector, is also an optimal solution.

Combining these two observations means that problem (7), which is formulated for any input $z$, can be rewritten only for non-negative sorted inputs. Note that we can always invoke the above observations to recover a solution to the original problem by first solving problem (9) to find $w^*$ and then $\Pi^{-1} D^{-1} w^*$ is an optimal solution of problem (7). $\qquad \square$

### B.4. Proof of Lemma 4

We first state a slighlty extended version of Lemma 4

**Lemma 10** (Optimality conditions – Extended). *Let the objective function in* (9) *be* $f(w)$. *Assume* $z_1 \geq z_2 \geq z_3 \geq z_4 > 0$. *The solution* $w^*$ *must be in one of the following three cases.*[5]

- *2-sparse: in which case* $w^* = [z_1, z_2, 0, 0]$.

- *3-sparse: in which case* $w_4^* = 0$ *and* $w_{1:3}^*$ *satisfies that*

$$
\begin{aligned}
0 < w_1^* &= z_1 - \lambda w_2^* w_3^*, \\
0 < w_2^* &= z_2 - \lambda w_1^* w_3^*, \\
0 < w_3^* &= z_3 - \lambda w_1^* w_2^*.
\end{aligned}
$$

- *Dense: in which case* $w^* = [w_1^*, w_2^*, w_3^*, w_4^*] > 0$ *satisfies that*

$$
w_i^* = z_i - \lambda \cdot \sum_{\{j_1, j_2\} \subset \{j \in [4], j \neq i\}} w_{j_1}^* w_{j_2}^*.
$$

*Moreover if* $z_1, z_2 > 0$,

1. $\lambda \geq z_3/(z_1 z_2)$ *is a necessary condition for the solution to be in the 2-sparse regime.*

2. $\lambda \geq z_4/(w_1^* w_2^* + w_2^* w_3^* + w_1^* w_3^*)$ *is a necessary condition for the solution to be in the 3-sparse regime, for* $w_{1:3}^*$ *to be the solution to* (11). *A weaker necessary condition is* $\lambda \geq z_4/(z_1 z_2 + z_2 z_3 + z_1 z_3)$.

*Proof.* If an optimal solution $w^*$ of problem (9) is 2-sparse, it means that $r_{2:4}(w^*) \equiv 0$. Therefore, $w^*$ is a minimizer of the quadratic term. Moreover, since $w^*$ is sorted in a descent order it means that $w_1^*, w_2^* > 0$ and $w_3^* = w_4^* = 0$. Therefore, the desired result immediately follows.

If an optimal solution $w^*$ of problem (9) is 3-sparse, it means that $w_4^* = 0$. Therefore, $w^*$ is also an optimal solution of problem (11). Moreover, since $w^*$ is a non-negative 3-sparse vector it follows that $w_1^*, w_2^*, w_3^* > 0$. This implies that the

---

[5]If we have equality in some of the inputs, i.e., $z_i = z_j$, permuting those two inputs also leads to valid solutions. We do not give precedence to either of those solutions but simply chose the one that follows the order provided by the sorting algorithm. For the proof we ignore this degeneracy to keep it simple.

non-negative constraint of problem (11) is inactive and therefore an optimal solution must be a stationary point of the objective function. The desired result follows by zeroing the gradient of the objective function of problem (11).

If an optimal solution $w^*$ of problem (9) is dense, it follows that $w^* > 0$. Then, the non-negative constraint of problem (9) is inactive at $w^*$. Therefore, $w^*$ must be a stationary point of the objective function of problem (9).

Since the optimization problem (9) is a minimization of the (non-convex) differentiable objective function $f$ over the non-negative constraint, it is well known that the corresponding KKT conditions are necessary for critical points $w^*$, which are compactly given by $\nabla f(w^*) \geq 0$ , $w^* \geq 0$ and $\nabla f(w^*)^T w^* = 0$. Therefore, from the third condition it follows that for any positive element of $w^*$ the corresponding partial derivative of $f$ must be zero. Moreover, by writing the gradient of the objective function of problem (9), we get that

$$\nabla f(w^*)^T w^* = (w^* - z + \lambda \nabla r_{2:4}(w^*))^T w^* = \|w^*\|^2 - z^T w^* + 3\lambda r_{2:4}(w^*),$$

where the last equality follows from the fact that $\nabla r_{2:4}(w)^T w = 3r_{2:4}(w)$ for any $w \in \mathbb{R}^4$. From here we immediately see that $z \geq w^*$ and if $r_{2:4}(w^*) \neq 0$ then $\lambda = (z^T w^* - \|w^*\|^2)/3r_{2:4}(w^*)$. From the first condition, that is $\nabla f(w^*) \geq 0$, we get that $\lambda \geq (z_i - w_i^*)/\nabla_i r_{2:4}(w^*)$ for all $1 \leq i \leq 4$ that $\nabla_i r_{2:4}(w^*) \neq 0$.

Since the optimal solutions must be monotonically decreasing in $i$ there are no other cases.

The two other necessary conditions about $\lambda$ stem from the KKT conditions of a critical point (with active constraints) for the constrained optimization problem. Specifically, the Lagrangian of $f$ $\mathcal{L}(w, \nu) = f(w) - \nu^T w$.

The conditions for critical points (including those at 0) are that there exists $\nu \in \mathbb{R}^4$ such that

1. Stationarity: $\nabla_w \mathcal{L}(w, \nu) = \nabla f(w) - \nu = 0$

2. Complementary slackness: $\nu_i w_i = 0$ for $i = 1, 2, 3, 4$.

3. Primal feasibility: $w \geq 0$

4. Dual feasibility: $\nu \geq 0$.

For any critical point to be 2-sparse i.e., $w_1 > 0, w_2 > 0, w_3 = w_4 = 0$, it needs to satisfy complementary slackness which implies $\nu_1 = \nu_2 = 0$. Moreover, stationarity condition gives $w_1 = z_1, w_2 = z_2$,

$$\nu_3 = \frac{\partial}{\partial w_3} f(w) = w_3 - z_3 + \lambda w_1 w_2 = -z_3 + \lambda z_1 z_2$$

and

$$\nu_4 = \frac{\partial}{\partial w_4} f(w) = -z_4 + \lambda z_1 z_2$$

Now the dual feasiblity condition $\nu_3, \nu_4 \geq 0$ gives the condition that $\lambda \geq z_3/(z_1 z_2)$.

Similarly, for any critical point to be 3-sparse, i.e., $w_1 > 0, w_2 > 0, w_3 > 0, w_4 = 0$, we have $\nu_1 = \nu_2 = \nu_3 = 0$, and that there exists

$$\nu_4 = \frac{\partial}{\partial w_4} f(w) = -z_4 + \lambda(w_1 w_2 + w_2 w_3 + w_3 w_1) \geq 0.$$

$w_1, w_2, w_3$ can be solved by the following non-linear system of equation

$$\begin{cases} w_1 + \lambda w_2 w_3 = z_1, \\ w_2 + \lambda w_1 w_3 = z_2, \\ w_3 + \lambda w_1 w_2 = z_3. \end{cases} \tag{22}$$

For this reason, if we are able to first identify a solution to (22) then verify that the $\nu_4 \geq 0$, then we certified that this solution is a critical point — a necessary condition for this sparse solution to be a global optimal solution. $\quad\square$

**B.5. Proof of Lemma 5**

*Proof.*

$$\nabla^2 g(w) = \begin{bmatrix} 1 & \lambda w_3 & \lambda w_2 \\ \lambda w_3 & 1 & \lambda w_1 \\ \lambda w_2 & \lambda w_1 & 1 \end{bmatrix} \quad \text{and} \tag{23}$$

$$\nabla^2 f(w) = \begin{bmatrix} 1 & \lambda(w_3 + w_4) & \lambda(w_2 + w_4) & \lambda(w_2 + w_3) \\ \lambda(w_3 + w_4) & 1 & \lambda(w_1 + w_4) & \lambda(w_1 + w_3) \\ \lambda(w_2 + w_4) & \lambda(w_1 + w_4) & 1 & \lambda(w_1 + w_2) \\ \lambda(w_2 + w_3) & \lambda(w_1 + w_3) & \lambda(w_1 + w_2) & 1 \end{bmatrix}. \tag{24}$$

Note that the conditions that $\nabla^2 g \succeq 0$ and $\nabla^2 f \succeq 0$ are *Linear Matrix Inequalities*, thus convex. $\qquad \square$

**B.6. Proof of Corollary 6**

*Proof.* It suffices to check the definition of a convex set, i.e., for any pair of local minima, their convex combination is also a local minimum.

Let $\mathcal{C}_4 := \{w \in \mathbb{R}^4 \mid \nabla^2 f(w) \succeq 0\}$ as in Lemma 5, where we showed that it is convex. Let $u^*, v^* \in \mathbb{R}^4$ be local minima of $f$. Then, since $u^*, v^* \in \mathcal{C}_4$ and $\mathcal{C}_4$ is convex, for any $0 \leq t \leq 1$, $f$ is convex at $w(t) := tu^* + (1 - t)v^*$.

By the definition of $\mathcal{C}_4$ when restricting to this set, $f$ is a convex function. By the first order definition of convex function

$$f(u^*) \geq f(v^*) + (u^* - v^*)^T \nabla f(v^*) = f(v^*)$$

similarly

$$f(v^*) \geq f(u^*) + (v^* - u^*)^T \nabla f(u^*) = f(u^*)$$

thus $f(u^*) = f(v^*)$.

By convexity of $f$ on the line segment, $f(w(t)) \leq tf(u^*) + (1 - t)f(v^*) = f(u^*)$ for all $t$. On the other hand, as $t \to 0$, we also have $f(w(t)) \geq f(u^*)$ due to that $u^*$ is a local minima. These two conditions imply that $f(w(t)) = f(u^*)$ for all $t \in [0, 1]$, i.e., $w(t)$ is also a local minimum.

To prove the second part we first show that the minimum of $f$ over $\mathbb{R}_+^4$ exists. Therefore note that for any $w \in \mathbb{R}_+^4$ if $w_i > z_i$, we have $\partial_i f(w) < 0$ for all $i \in [4]$, implying that $f$ is minimized within $w_i \leq z_i$ which is a closed set. Hence there exists $w^* \in \mathbb{R}_+^4$ such that $w_i^* \leq z_i$ and $f(w^*) = \min_{w \in \mathbb{R}_+^4} f(w)$. Now assume $w_i^* > 0$ for all $i \in [4]$, then $w^*$ necessarily is a local minimum and $\nabla^2 f(w^*) \succeq 0$. Conversely, since all local minima of a convex function attain the same value, the statement follows.

The statements for $g$ follow in analogy. $\qquad \square$

**B.7. Proof of Theorem 7**

*Proof.* The optimal solution to (9) has three possibilities: dense, 3-sparse, 2-sparse.

If the solution to (9) is 2-sparse, $r_{2:4}(w) = 0$ then $w^{(2)}$ is the solution due to that it minimizes $\frac{1}{2}\|w - z\|^2$ subject to the 2-sparse constraint.

If the solution to (9) is 3-sparse with $w_{1:3} > 0$, then $w_{1:3}$ is also the solution to (11), moreover $\nabla^2 g(w_{1:3}) \succeq 0$. Since the constraint is not active, and $w_{1:3}$ is in the strict interior of the constraint, thus $\nabla g(w_{1:3}) = 0$. Since $w_{1:3}$ is feasible, it is also optimal for (13). Any other solution $\tilde{w}_{1:3}$ to (13) (if exists) must satisfy $g(\tilde{w}_{1:3}) = g(w_{1:3})$ thus is also an optimal solution to (11) and (9).

Similarly, if the solution to (9) is dense with $w > 0$ (for all coordinates), then $\nabla^2 f(w) \succeq 0$ and $\nabla f(w) = 0$ (due to stationarity and complementary slackness). This checks the feasibility of $w$ in (14) which implies that $w$ is an optimal solution to (14). Let $\tilde{w} \in \mathbb{R}^4$ be any other optimal solution to (14), $f(\tilde{w}) = f(w)$, thus $\tilde{w}$ is also an optimal solution to (9).

To conclude, in each of the three cases, Line 2-4 of the algorithm returns the optimal solution. $\qquad \square$

### B.8. Proof of Fact 8

*Proof.* a) At $w = [0, 0, 0, 0]$ the Hessian (24) is just the identity and hence positive.

b) First note that as soon as one off-diagonal entry of (24) is larger than one, the Hessian is not positive semidefinite anymore. Thus positivity of the Hessian implies that all off-diagonal entries are less or equal to 1. We can then use Gershgorin disc theorem and obtain that all eigenvalues are between $[-2, 4]$, hence $\gamma_{\max} \leq 4$. □

## C. Other Regularizers for 2:4 Sparsity

Our main theoretical contribution is to show that the non-convex proximal operator can be solved efficiently by decomposing it into smaller subproblems. In principle, there also exist simpler regularizers that induce 2:4 sparsity and whose proximal operator can be solved more efficiently. Here we introduce three further proximal operators of simpler regularizers. To be concise, we assume Lemma 3 has already been applied, i.e., $w_1 \geq w_2 \geq w_3 \geq w_4 \geq 0$ and we define the regularizers

$$R_0(w_1, w_2, w_3, w_4) := \begin{cases} 0 & \text{if } w_4 = w_3 = 0, \\ 1 & \text{if } w_4 = 0, w_3 > 0, \\ 2 & \text{if } w_4 > 0. \end{cases} \tag{25}$$

$$R_1(w_1, w_2, w_3, w_4) := w_4 + w_3, \tag{26}$$

$$R_2(w_1, w_2, w_3, w_4) := \frac{1}{2}(w_4^2 + w_3^2). \tag{27}$$

Each of those regularizers equals 0 if and only if the four weights have a 2:4 sparse pattern. So those regularizers do also enforce 2:4 sparsity. Let's look at their proximal operators – again assuming $z_1 \geq z_2 \geq z_3 \geq z_4 \geq 0$.

$$\arg\min_{w \in \mathbb{R}_+^4} f(w), \quad \text{with} \tag{28}$$

$$f(w) := \frac{1}{2}\|w - z\|^2 + \lambda R_p(w_1, w_2, w_3, w_4). \tag{29}$$

Note that the objective of those proximal operators can all be easily be decomposed for each of the variables and trivially $w_1^* = z_1$ and $w_2^* = z_2$. The other optimizations are also simple. For $R_0$ we have $w_3^* = 0$ if $\lambda > \frac{1}{2}z_3^2$ else $z_3$; for $R_1$ we have $w_3^* = \max(z_3 - \lambda, 0)$; For $R_2$ we have $w_3^* = z_3/(1 + \lambda)$. For $w_4^*$ it follows in analogy for all three cases.

Thus, depending on $\lambda$, $R_0$ forces weights to be zero if they are below a threshold (hard thresholding), $R_1$ does a soft-thresholding, and $R_2$ leads to a shrinkage. For large but finite $\lambda$, $R_0$ and $R_1$ lead to exact $2:4$ sparsity, whilst $R_2$ does not lead to exact 2:4 sparsity for finite $\lambda$.

However, those regularizers commit very early to the sparsity pattern, and none of the behavior of Figure 1 are observed for them. Furthermore, on the toy problem in Section 4.1 they cannot find the optimal solution. We hence focused the main part of this work on the more elaborate proximal operator. For completeness we also report perplexity numbers for it in Table 2 as well as the runtime. Since the optimization problems are trivial and have a close form solution, they are significantly faster, but lead to slightly worse perplexity.

## D. Experiment Details

In Figures 3, 4, and 5 we show further ablations and insights into our main experiments. Table 2 shows the different runtime of the pruning methods and Table 3 shows results of the pruned models on downstream tasks.

### D.1. Using the Hessian of the Unpruned Model

For our studies in the main paper we compute the Hessian matrix for each linear layer once before the pruning. This allows us to prune layers in parallel and distribute the pruning on multiple GPUs, because the local Hessians do not depend on each other. The alternative is to prune the matrices sequentially and propagate the pruned inputs. In our ablation Table 4 we find however, that it's effect is overall very minor, and it is not structurally better than using the unpruned inputs.

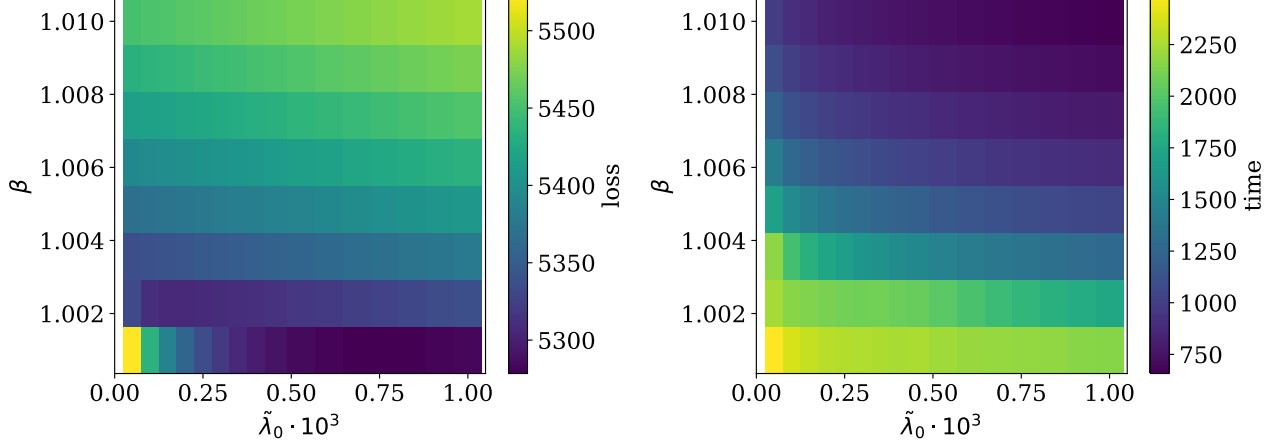

*Figure 3.* Ablation for the regularizer schedule Equation (15) on the down projection of layer 18 of Llama-3.1 8B.

*Table 2.* Wall clock runtime of different pruning approaches on Llama-3.1 8B Instruct. The proposed regularizer leads to the best perplexity, however, also incurs the highest pruning cost. Since this is a once-off effort, and negligibly small compared to pretraining costs, this can be disregarded in production settings.

|  | Runtime [s] | Runtime [min] | C4 Perplexity | Wiki Perplexity |
|---|---|---|---|---|
| *proposed regularizer [Section 3.2]* | 11,170 | 186.17 | **26.44** | **20.15** |
| *L0 regularizer, Equation (25)* | 1,346 | 22.43 | 27.25 | 20.21 |
| *L1 regularizer, Equation (26)* | 1,242 | 20.70 | 28.46 | 20.70 |
| *L2 regularizer, Equation (27)* | 2,628 | 43.80 | 27.84 | 20.43 |
| wanda | **258** | 4.30 | 43.30 | 26.67 |
| *wanda+GD* | 1,000 | 16.67 | 28.45 | 20.71 |
| sparsegpt | 313 | 5.22 | 35.55 | 26.19 |
| *sparsegpt+GD* | 1,064 | 17.73 | 27.54 | 20.85 |

*Table 3.* Accuracy Evaluation of pruned versions of Llama-3.1 8B Instruct on 6 Downstream Tasks. The methods (*+GD*) introduced in this work reduce the accuracy gap by around 3% to 5% against previous methods.

| model | Mean | MMLU | GSM8K | Winogrande | Hellaswag | TruthfulQA | ai2_arc |
|---|---|---|---|---|---|---|---|
| original | 64.28% | 67.70% | 75.36% | 73.79% | 59.19% | 37.58% | 72.04% |
| wanda | 35.78% | 37.02% | 2.50% | 62.75% | 38.94% | 26.19% | 47.27% |
| *wanda + GD* | 40.76% | 40.65% | 11.83% | 65.27% | 43.98% | 27.42% | 55.44% |
| sparsegpt | 37.59% | 34.97% | 6.07% | 64.56% | 41.50% | 24.60% | 53.86% |
| *sparsegpt + GD* | 40.98% | 42.94% | 8.95% | 66.06% | 44.74% | 27.17% | 56.00% |
| DSnoT | 35.01% | 36.98% | 1.36% | 59.75% | 36.71% | 26.32% | 48.96% |
| *prox + GD* | 40.81% | 41.82% | 10.01% | 65.51% | 44.82% | 27.29% | 55.41% |

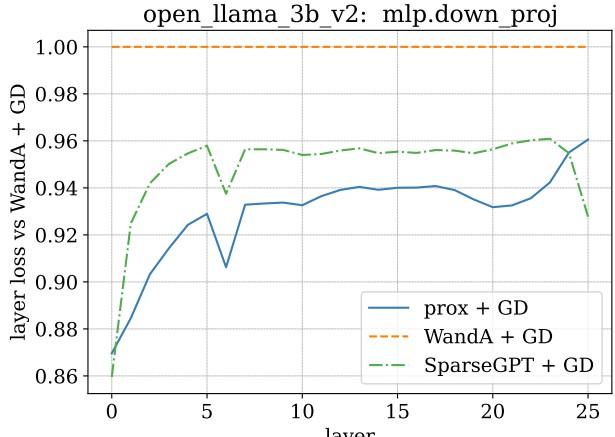

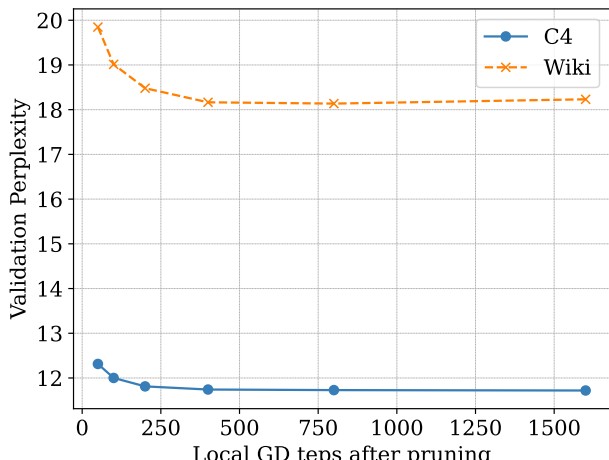

*Figure 4.* Layerwise local squared loss on an example matrix of openLlama 3bv2. We show the loss relative to the loss WandA (with masked gradient updates) incurrs. As intended by design, the proximal approach of pruning leads to smaller local squared loss. In Table 1 we see that this also translates to a better end to end performance in terms of perplexity.

*Figure 5.* Effect of number of GD steps Equation (5) after masking with Wanda. The perplexity converges quickly and the used 1000 steps in the main paper suffice to drive the local optimization to convergence. This shows that the proximal operator indeed identifies a more suitable mask.

*Table 4.* The effect of propagating the pruning errors is negligible.

| Model | Method | Hessian | C4 | Wiki |
|-------|--------|---------|--------|-------|
| 3b_v2 | dense | | 9.68 | 14.15 |
| | wanda | Original | 29.49 | 60.35 |
| | wanda | Pruned | **28.65** | **55.98** |
| 7b_v2 | dense | | 8.84 | 12.15 |
| | wanda | Original | 17.61 | 27.00 |
| | wanda | Pruned | **17.59** | **26.88** |
| 13b | dense | | 8.11 | 11.52 |
| | wanda | Original | **13.17** | **35.39** |
| | wanda | Pruned | 13.30 | 37.61 |

