# OpenReview forum: "A Proximal Operator for Inducing 2:4-Sparsity"
_ICML.cc/2025/Conference — Submitted to ICML 2025_

### Official Review · Reviewer_aPhy · 2025-02-26

**Overall Recommendation:** 3

**Summary:**

This paper proposes to get 2:4 weight sparsity gradually using the proximal gradient method on a per-matrix level for pruning large language models (LLMs) after pretraining.
1) Firstly, a special regularizer with 2:4 sparsity null space is proposed, and the authors show that we can control the structured sparsity of its corresponding proximal operator continuously by changing a hyper-parameter $\lambda$, both theoretically (see Lemma 10) and experimentally (see Figure 1).
2) Secondly, the authors propose to solve the proposed non-convex proximal operator efficiently by dividing it into three convex subproblems (see Theorem 7), based on the optimal point classification lemma (see Lemma 4) and the convexity of the space composed of points with positive semi-definite hessian matrices (see Lemma 5).
3) Thirdly, the masked-gradient update is proposed to partially compensate for the pruning loss, which can be adopted by all the methods for post-training pruning.
4) Lastly, experiments in toy settings and real LLM pruning tasks demonstrate the proposed method can achieve state of the art performance.

**Claims And Evidence:**

The claims made in the submission are supported by clear and convincing evidence, though the improvement seems marginal in LLM pruning experiments.

**Essential References Not Discussed:**

None.

**Experimental Designs Or Analyses:**

The experimental designs are sound in general, yet with some omissions:
1) In table 1, results of prox without GD do not provided.
2) There is no ablation study on the number of calibration samples (see Table 17 in [1]).

[1] A simple and effective pruning approach for large language models. In ICLR, 2024.

**Methods And Evaluation Criteria:**

The proposed method is instructive for post-training structured sparsity, though the application of it is limited to N:M sparsity for N < 3.

**Other Comments Or Suggestions:**

None.

**Other Strengths And Weaknesses:**

1) Table 1 shows that the performance improvement achieved by proposed method is marginal on models with fewer than 8B parameters, and does not exist on models with more than 8B parameters.
2) The masked-gradient update seems useless for wanda when pruning models with more than 8B parameters (see the validation perplexity on Wiki in Table 1).
3) The proposed method is more time-consuming than the previous approaches. Further, it requires a extra hyper-parametric scheduling and there is no theory to guide its implementation.
4) This paper is well-written and easy to follow, though in my view, Section 3.3 (Solution of the 2:4-Proximal Operator) needs to be reorganized for better understanding (see Theoretical Claims).
5) There are some inappropriate marks and layouts used in this paper: $w\in W$ in Equation (8) (see line 190) is incompatible with Equation (6) (see line 210); the position of Figure 1 is inappropriate since it appears on the top of page 2 and is quoted by Section 3 on page 4.

**Questions For Authors:**

1) Why do you adopt exponential schedules for $\lambda$ instead of others?
2) Figure 2 shows that the proposed method is more effective when the correlation between input features is high. What does such correlation look like for LLMs? Is there any real-world task to prove your proposed method has a prominent advantage?

**Relation To Broader Scientific Literature:**

This paper explores regularizers for 2:4 sparsity, and develops a special proximal gradient method to enhance the sparsity of weights gradually. The squared loss can be found in formula (1) of [2].

[2] Sparsegpt: Massive language models can be accurately pruned in one-shot. In ICML, 2023.

**Theoretical Claims:**

1) The proofs of Theorem 1, Fact2, Lemma3-5 are correct. Corollary 6 seems trivial and a little confusing, because the authors do not clearly point out which targets have linear constraints. A better implementation of Algorithm 1 may be reduce dimension when negative numbers appear during the gradient descent, according the technique used to prove Lemma 3. I do not check the correctness of Fact 8, and the rationality of Conjecture 9.
2) There is a minor mistake in line 748: $\partial_if(w)<0$ is false, it should be $\partial_if(w)>0$, and this does not affect the validity of the whole proof.

---

> ### Author Rebuttal · Authors · 2025-03-27
>
> Thank you very much for thoroughly checking our theoretical claims and helping to improve our presentation. We will carefully take your considerations into account and will revise Section 3.3 around Corollary 6 to ensure it is easier to follow.
>
> We want to briefly reply to some of your comments and hope that this helps for clarification. In particular we ran a new ablation study on the relevance of calibration samples that we will add to the paper.
>
> > There is no ablation study on the number of calibration samples
>
> Thank you, this is an important point and was also raised by reviewer 5gi2. We ran a similar ablation study to wanda on the 3B model.  Except when using only a single sample, our proposed prox+GD outperforms all other methods when using the same amount of calibration data. Please check our reply to 5gi2 for the full results.
>
> > In table 1, results of prox without GD do not provided.
>
> Indeed this is on purpose, we do not want to propose prox without masked gradient as a method. This would mean that the optimization stops as soon as we have a 2:4 mask, without allowing enough iterations for the remaining weights to optimally adjust to this mask.
>
> >Why do you adopt exponential schedules for instead of others?
>
> First note (Section 3.2) that having a regularizer that grows exponentially guarantees that the regularizer eventually is large enough such that the solution to the proximal operator will be exactly 2:4 sparse, which we need to ensure finite runtime. In other words: without making assumptions about the problem space, the regularizer eventually needs to be scheduled towards infinity. Furthermore, as we can empirically see from Figure 5 in the appendix our exponential schedule leads to a good balance of runtime and accuracy. We believe that at the initial steps of the algorithm it is important to not increase the regularizer too quickly, as this results in committing to a certain sparsity pattern too early. On the other hand once the sparsity pattern is mostly established  we can drive the sparsification faster. This is well done by an exponential schedule.
>
> > Figure 2 shows that the proposed method is more effective when the correlation between input features is high. What does such correlation look like for LLMs? Is there any real-world task to prove your proposed method has a prominent advantage?
>
> Please also see our reply to reviewer fgi2 ("Performance on 70B scale"). We are in fact struggling to fully nail down what exactly it is that changes between the model scales. On small models using off-diagonal information seems to be relevant, whilst on large models it does not help significantly. Whilst currently the research community (and us) are focusing on pruning open source Llama models, it is conceivable that on other future models, the off-diagonal information will again be more relevant even at larger scales.

---

> > ### Comment · Reviewer_aPhy · 2025-04-02
> >
> > Thanks for the reply. The ablation study on the number of calibration samples for pruning the 3B model is useful, and your explanation of using exponential schedules makes sense.
> >
> > I agree that treat the prox + masked gradient update as a complete method, yet I still think that prox without GD, or prox with different GD steps can be an important ablation experiment for your paper.
> >
> > To better characterize the properties of the matrix from real-world LLMs and analyse the effectiveness of your proposed method, I think references on Basis Pursuit for Compressed Sensing will be helpful. You can also check the Kruskal Rank, Mutual Coherence, Restricted Isometry Property of LLM matrices with different sizes.

---

> > > ### Author Response · Authors · 2025-04-03
> > >
> > > Thank you for the timely reply and the opportunity to follow up once more.
> > >
> > > > I still think that prox without GD, or prox with different GD steps can be an important ablation experiment for your paper.
> > >
> > > Indeed as an ablation to gain further insights and understanding of the prox+GD method, it helps to run an ablation for the GD steps impact also for prox. We did run this for OpenLlama3B now as well and present the results in the table and created a plot here https://ibb.co/Nnd8JMhN . Aside from the number of GD steps, the setup is identical to the one used in the main paper.
> > > Our main observations: First, even with zero GD steps after masking, prox outperforms wanda with 1000 GD steps (c4: 18.23, wiki: 33.05 | Table 1 of submission) . This confirms that prox indeed finds a better mask for this model. Second, we observe that the masked gradient steps further improve the perplexities. However, it quite quickly converges. These finding is aligned with the GD steps ablations we did for wanda on the 13B model (see Figure 5).
> > >
> > > We will streamline those additional ablations in the final version of the paper and discuss the insights in the main part.
> > >
> > > | steps after masking |     0 |    50 |   100 |   200 |   400 |   800 |  1000 |  1600 |
> > > |---------------------|------:|------:|------:|------:|------:|------:|------:|------:|
> > > | c4                  | 16.92 | 16.70 | 16.56 | 16.41 | 16.31 | 16.27 | 16.27 | 16.26 |
> > > | wiki                | 30.63 | 29.99 | 29.58 | 29.12 | 28.84 | 28.66 | 28.62 | 28.63 |
> > >
> > > > To better characterize the properties of the matrix from real-world LLMs and analyse the effectiveness of your proposed method, I think references on Basis Pursuit for Compressed Sensing will be helpful. You can also check the Kruskal Rank, Mutual Coherence, Restricted Isometry Property of LLM matrices with different sizes.
> > >
> > > Thank you the additional pointers. We already tried to gain further understanding by looking at the effective rank for example or the relation of weight mass on the diagonal vs off-diagonal (of the Hessian). We will look into the additional references you provided and will include them in the discussion. We hope to gain some further insights here -- again those are equally important to understand the existing methods Wanda and sparsegpt. However, at the time being, we can of course not *promise* anything in this regard.
> > >
> > > One simple hypothesis we have ruled out is that the Hessian is more diagonal in larger models. If that were the case, Theorem 1 would imply that Wanda performs well and that GD steps have limited effect. However, our empirical measurements do not support this—the Hessian is not noticeably more diagonal in larger models than in smaller ones. We believe that the true cause might be hidden in the interplay of weight and activations, which makes the analysis more challenging.
> > >
> > >
> > > Thank you for the constructive feedback—we hope this clarifies our contributions and that you will consider supporting our paper.

---

### Official Review · Reviewer_fybz · 2025-03-06

**Overall Recommendation:** 4

**Summary:**

The proposed method  solves the complex proximal operator for 2:4 sparsity by optimized masked gradient updating. The theoretical analysis provides clear support for the mechanism of the proposed method. The authors have conduct extensive experiments to validate the effectivenss of the proposed method.

**Claims And Evidence:**

Yes, the authors provide clear therotical and experimental results to prove their claim.

**Essential References Not Discussed:**

No. All nessasry related work are cited and discussed.

**Experimental Designs Or Analyses:**

The experimental designs and analyses are sound. The authors provide both toy experiments to illustrate the mechanism of the propose method and extensive comparative results against existing LLM pruning approaches.

**Methods And Evaluation Criteria:**

The authors benchmark the proposed method for sparsifying representative LLMs, particularly demonstrating its effectiveness in improving the performance of exisiting 2:4 sparsity methods. The evaluation criteria is generally sound. The authors further provide more context on speedups, with code showcases 2:4 sparsity benchmark.

**Other Comments Or Suggestions:**

It would be better to show some real speedup for 2:4 sparsity in LLMs.

**Other Strengths And Weaknesses:**

Strengths:

* This paper presents a complete review of the proximal operator's solution, demonstrating its effective resolution in the context of 2:4 sparsity, providing substantial theoretical support for future studies.
* The method can be directly applied to large language model compression and optimization in resource-constrained environments, considerably enhancing inference efficiency.

Weaknesses:

*  Although the proposed method is theoretically valid, its complexity may cause computational and implementation burdens, particularly with LLMs.

**Questions For Authors:**

Minor typo: Is there some specific reasons to use 'WandA' instead of 'Wanda'?

**Relation To Broader Scientific Literature:**

This paper offers fresh insights for 2:4-Sparsity, particularly for its application to LLMs.

**Theoretical Claims:**

I have checked the correctness of all proofs for theoretical claims.

---

> ### Author Rebuttal · Authors · 2025-03-27
>
> Thank you very much for checking the correctness of all proofs for our theoretical claims and your positive assessment. We want to briefly answer two comments:
>
> > It would be better to show some real speedup for 2:4 sparsity in LLMs.
>
> We kindly ask you to check our response to reviewer 8YCw where we provide more context on speedups -- the code for our benchmark is attached below. We believe that realizing the full speedup of 2:4 is an important effort orthogonal to our contribution which focuses on the accuracy.
>
> > Minor typo: Is there some specific reasons to use 'WandA' instead of 'Wanda'?
>
> Yes, we thought it is more instructive to use **W**eights **and** **A**ctivations as abbreviation. However, we realized that the authors themselves used Wanda / wanda. So we will change it to their notation for consistency.
>
> We are happy to answer any further questions that might come up.
>
> ---
> ---
> ### 2:4 sparsity benchmark with pytorch
> ```
> import pandas as pd
> import torch
> import time
> from torch.sparse import to_sparse_semi_structured
>
> shapes = {
>     "8b_qkv_proj": (4096 + 1024 + 1024, 4096), # assuming fused QKV
>     "8b_o_proj": (4096, 4096),
>     "8b_up_gate_proj": (14336 * 2, 4096), # assuming fused Up/Gate
>     "8b_down_proj": (4096, 14336),
>
>     "70b_qkv_proj": (8192 + 1024 + 1024, 8192), # assuming fused QKV
>     "70b_o_proj": (8192, 8192),
>     "70b_up_gate_proj": (28672 * 2, 8192), # assuming fused Up/Gate
>     "70b_down_proj": (8192, 28672),
> }
>
> REPEATS = 10
> WARMUPS = 10
> BATCH_SIZES = [1, 16, 32, 8192, 8192*2, 8192*4]
> results_df = pd.DataFrame(columns=["Layer", "Batch Size", "Speedup"])
> with torch.no_grad():
>     for matrix in shapes:
>         for bs in BATCH_SIZES:
>             out_featues, in_features = shapes[matrix]
>
>             # Create input and dense weight
>             dense_weight = torch.Tensor([0, 0, 1, 1]).tile((out_featues, in_features//4)).half().cuda()
>             x = torch.rand(in_features, bs).half().cuda()
>             sparse_weight = to_sparse_semi_structured(dense_weight)
>
>             # Function to benchmark
>             def benchmark_matmul(weight, x):
>                 total_time = 0
>                 torch.cuda.synchronize()
>                 for _ in range(WARMUPS):
>                     y = torch.mm(weight,x)
>
>                 for _ in range(REPEATS):
>                     torch.cuda.synchronize()
>                     start = time.time()
>                     # torch.cuda.synchronize()
>                     y = torch.mm(weight,x)
>                     torch.cuda.synchronize()
>                     end = time.time()
>                     total_time += (end - start)
>                 return total_time / REPEATS
>
>             # Run benchmarks
>             dense_time = benchmark_matmul(dense_weight, x)
>             sparse_time = benchmark_matmul(sparse_weight, x)
>
>             # print(f"Dense time per run: {dense_time * 1000:.3f} ms")
>             # print(f"Sparse time per run: {sparse_time * 1000:.3f} ms")
>
>             print(f"Matrix: {matrix}, bs: {bs}, Speedup: {dense_time / sparse_time:.2f}x")
>             results_df = pd.concat([results_df, pd.DataFrame({"Layer": matrix, "Batch Size": bs, "Speedup": dense_time / sparse_time}, index=[0])], ignore_index=True)
>
> results_df.to_csv("sparse_matmult_results.csv", index=False)
> ```

---

> > ### Comment · Reviewer_fybz · 2025-04-02
> >
> > I acknowledge and appreciate the authors' efforts in providing the speedup results of 2:4 sparsity.  I also agree with the authors that practical speedup doesn't constitute a key limitation to this individual paper, which requires collective progress across the field.  I maintain my positive assessment of this work.

---

> > > ### Author Response · Authors · 2025-04-07
> > >
> > > Thank you for the explicit response to our rebuttal and for the positive assessment.

---

### Official Review · Reviewer_8YCw · 2025-03-07

**Overall Recommendation:** 3

**Summary:**

This paper proposes a proximal operator to improve the one-shot N:M weight pruning for large language models. The paper finds better sparsity masks in trained models by minimizing a regularizer jointly with local squared loss though deriving the proximal operator. Besides the algorithm for better masks, the paper introduces masked gradient updates to further minimize the local squared loss given a mask. These techniques can improve existing pruning methods with 2:4 sparsity on up to 13B models. On 70B models, the performance is on par. The authors also illustrate their method on toy problems.

**Claims And Evidence:**

The claims on the effectiveness of gradient descent and proximal operator made in the submission are supported by clear and convincing evidence

**Essential References Not Discussed:**

I didn’t identify essential references not discussed in the paper.

**Experimental Designs Or Analyses:**

I have checked the validity of the experimental designs and analyses.
Experiments are valid:

1. Toy experiments in Section 4.1 illustrate that prox can better utilize the correlation between input features.

2. Table 1 tests the dense models, state of the art sparse models with and without gradient updates, and prox with gradient updates. The result can support the claims: “On models up to 13B we improve over previous state of the art algorithms, whilst on 70B models we match their performance”.

**Methods And Evaluation Criteria:**

The proposed methods and evaluation criteria make sense for the problem: inducing 2:4-sparsity in a one-shot manner and evaluating the perplexity.

**Other Comments Or Suggestions:**

I don’t have other comments or suggestions.

**Other Strengths And Weaknesses:**

The proximal operator is effective for small and medium models but there is no comparison of the inference latency between dense and sparse models.

**Questions For Authors:**

Are there better optimization objectives than the squared loss (Equation (1)) given that perplexity is the ultimate objective?

**Relation To Broader Scientific Literature:**

Although 2:4-sparsity is an important hardware feature provided by NVIDIA GPUs, it is unclear how to tailor LLMs to such structural sparsity without sacrificing model performance. This paper proposes an effective method for inducing 2:4-sparsity.

**Theoretical Claims:**

I have checked the correctness of proofs for these theoretical claims:

Proof B.1. for Theorem 1

Proof B.2. for Fact 2

Proof B.3. for Lemma 3

Proof B.4. for Lemma 4

Proof B.5. for Lemma 5

Proof B.6. for Corollary 6

Proof B.7. for Theorem 7

Proof B.8. for Fact 8

I also did not find counterexamples for Conjecture 9

---

> ### Author Rebuttal · Authors · 2025-03-27
>
> Thank you very much for checking all our theoretical statements and the review of our paper. Below we provide a discussion on other optimization objectives as suggested in your review and some data and considerations on speedups.
>
> ### Optimization objectives beyond squared loss
> First we recall the motivation for using the squared loss. This builds on Hassibi & Stork (1992) "Second order derivatives for network
> pruning: Optimal Brain Surgeon", Section 2. If the Assumption A1: "the model is trained to convergence" holds, then by a Taylor expansion of the loss, a squared loss of the "global" hessian is the first relevant order.
> Note, however, that this global Hessian scales quadratically with the number of total parameters, which is completely impossible to handle for LLMs. Now we can make a second assumption A2: "suppose that the Hessian is *block-diagonal*, where each block contains the parameters of a single linear layer."
>
> Combining assumptions A1 & A2 we have that in first relevant order the *local squared loss* is the best objective to optimize -- which is what we consider. Furthermore, a linear squared loss allows for efficient optimization with theoretical guarantees. It would be non-trivial to (efficiently) apply our proximal optimization to other objectives.
>
> That said, neither assumptions A1 and A2 are fully met in practice.
>
> Recent work of Dong et al (ICML'24) "Pruner-Zero: Evolving Symbolic Pruning Metric from scratch for Large Language Models" makes a systematic study of which information is relevant for pruning. However, their approach requires significantly more computational resources, as it requires end-to-end gradient information.  Furthermore, they consider only a static pruning approach, i.e., no gradual weight updates during pruning. Unfortunately, their codebase does not support the llama-3 series of models and we were unable to compare directly.
>
> ### Speedup
> NVIDIA reports 1.1x to 1.2x speedups with 2:4 sparsity on ResNext-101 model (https://tinyurl.com/uhd2rhvb). However, we are not able to independently verify their results.
> Unfortunately, we were also unable to measure realized speedups using the current vLLM package with a 2:4 sparse checkpoint. We ran a Pytorch benchmark on an A100 GPU. Note that Pytorch support is in itself experimental. Notably, we observed significant slowdowns in many memory-bound workloads.
> However, we do see speedups up to 1.6x on some shapes which is consistent with what wanda reported (their Table 5). We anticipate that as the hardware and software stack matures, these speedups will improve.
>
> We hope this is not considered a shortcoming of our work. Fully realizing the speedup is an important effort orthogonal to our contribution.
>
> [Benchmark script in the response to reviewer fybz]
> | Layer            | Batch Size | Speedup |
> |------------------|------------|---------|
> | 8b_qkv_proj      | 1          | 0.07    |
> | 8b_qkv_proj      | 16         | 0.08    |
> | 8b_qkv_proj      | 32         | 0.08    |
> | 8b_qkv_proj      | 8192       | 0.92    |
> | 8b_qkv_proj      | 16384      | 1.22    |
> | 8b_qkv_proj      | 32768      | 1.32    |
> | 8b_o_proj        | 1          | 0.05    |
> | 8b_o_proj        | 16         | 0.06    |
> | 8b_o_proj        | 32         | 0.06    |
> | 8b_o_proj        | 8192       | 0.75    |
> | 8b_o_proj        | 16384      | 1.08    |
> | 8b_o_proj        | 32768      | 1.26    |
> | 8b_up_gate_proj  | 1          | 0.20    |
> | 8b_up_gate_proj  | 16         | 0.23    |
> | 8b_up_gate_proj  | 32         | 0.24    |
> | 8b_up_gate_proj  | 8192       | 0.83    |
> | 8b_up_gate_proj  | 16384      | 0.90    |
> | 8b_up_gate_proj  | 32768      | 0.88    |
> | 8b_down_proj     | 1          | 0.11    |
> | 8b_down_proj     | 16         | 0.12    |
> | 8b_down_proj     | 32         | 0.12    |
> | 8b_down_proj     | 8192       | 1.29    |
> | 8b_down_proj     | 16384      | 1.51    |
> | 8b_down_proj     | 32768      | 1.52    |
> | 70b_qkv_proj     | 1          | 0.15    |
> | 70b_qkv_proj     | 16         | 0.18    |
> | 70b_qkv_proj     | 32         | 0.18    |
> | 70b_qkv_proj     | 8192       | 1.28    |
> | 70b_qkv_proj     | 16384      | 1.41    |
> | 70b_qkv_proj     | 32768      | 1.47    |
> | 70b_o_proj       | 1          | 0.13    |
> | 70b_o_proj       | 16         | 0.15    |
> | 70b_o_proj       | 32         | 0.15    |
> | 70b_o_proj       | 8192       | 1.26    |
> | 70b_o_proj       | 16384      | 1.43    |
> | 70b_o_proj       | 32768      | 1.44    |
> | 70b_up_gate_proj | 1          | 0.53    |
> | 70b_up_gate_proj | 16         | 0.56    |
> | 70b_up_gate_proj | 32         | 0.56    |
> | 70b_up_gate_proj | 8192       | 0.91    |
> | 70b_up_gate_proj | 16384      | 0.90    |
> | 70b_up_gate_proj | 32768      | 0.93    |
> | 70b_down_proj    | 1          | 0.30    |
> | 70b_down_proj    | 16         | 0.32    |
> | 70b_down_proj    | 32         | 0.33    |
> | 70b_down_proj    | 8192       | 1.53    |
> | 70b_down_proj    | 16384      | 1.61    |
> | 70b_down_proj    | 32768      | 1.53    |

---

### Official Review · Reviewer_5gi2 · 2025-03-13

**Overall Recommendation:** 2

**Summary:**

The paper presents a post-training pruning method to induce N:M sparsity. The two main innovations of this article are:

1) It proposed a Gradient Descent approach that can be implemented to any post-training pruning method to compensate the pruning loss.

2) Combining with the proposed Regularization, this article is able to induce N:M sparsity gradually.

**Claims And Evidence:**

Most of the claims made in the submission are supported by clear and convincing evidence.

However, the author claims "Our approach will consume more compute and time, but this is well invested. Compared against the cost of training the models and the inference cost, the additional cost to find a better mask and model is well invested." But on large models, such as LLaMA3.1-70b, the proposed methods don't show an advantage against merely Wanda on wikitext2 datasets. Also, the running time of the proposed method on the large models is not reported.

**Essential References Not Discussed:**

The related works are well discussed in this article.

**Experimental Designs Or Analyses:**

I have checked the experimental designs

**Methods And Evaluation Criteria:**

The proposed methods make sense for the problem.

**Other Comments Or Suggestions:**

Please see the questions.

**Other Strengths And Weaknesses:**

**Weakness:**

1. The proposed method does not provide an advantage for large models. The authors should investigate why it fails to improve performance at larger scales.

2. The proposed method requires 1,000 calibration samples, which is significantly more than those used in other post-training pruning methods such as Wanda and SparseGPT. The paper should also examine how the number of calibration samples impacts the effectiveness of their approach.

3. My concern is that, despite using a large number of calibration samples and having a long runtime, the proposed method still fails to sufficiently bridge the performance gap caused by N:M pruning.

As the article said, the pruned 13B/7B models fall clearly behind the dense 7B/3B model. Therefore, I didn't see the potential of the proposed method.

**Questions For Authors:**

1. The article claims the proposed method "Consume more compute and time, but this is well invested." Could you also report the runtime for the 70B model?  .

2. What is the number of calibration samples used in SparseGPT and Wanda for comparison?

3. Could you perform a sensitivity analysis on the number of calibration samples to assess its impact on performance?

**Relation To Broader Scientific Literature:**

This article proposed a gradient descent method that can be implemented to all the existing post-training pruning methods.

**Theoretical Claims:**

I checked the theoretical claims and I didn't find mistakes.

---

> ### Author Rebuttal · Authors · 2025-03-27
>
> Thank you for the detailed review and confirming that our theoretical claims and the proposed method make sense. We understand that your reluctance comes from missing ablations on *calibration samples* as well as the unconvincing *performance* on the current 70B scale Llama models. We provide a new ablation on calibration samples below and discuss the 70B performance.
>
> We hope that our conceptual contributions as well as the strong empirical improvements over existing SOTA on smaller models and across calibration samples can offset your concerns on the 70B model. Please let us know if there is anything else we can elaborate on.
>
>
> ### **Ablation study on the effect of calibration samples:**
> First let us clarify that for our experiments in the submission we were using the exact same number of calibration samples for all methods (we will clarify this at the start of Section 4.3). Additionally we now ran a new ablation on the calibration samples for the 3B model (we consider the same number of samples as in Table 17 of the wanda paper).
> We make three main observations:
> - a/ wanda (which uses only the diagonal of the Hessian) converges quickly as known from prior work.
> - b/ methods that use masked gradient updates can consistently improve their validation perplexity when using more calibration data. However, with diminishing returns (note that the number of samples is increased exponentially here).
> - c/ Except when using only a single sample, our proposed prox+GD outperforms all other methods when using the same amount of calibration data.
>
> We are thankful for this suggestion and include those results in the updated paper, as they provide valuable further insights.
>
> | calibration samples |    1   |     1    |   16  |   16  |   32  |   32  |   64  |   64  |  128  |  128  |  256  |  256  |  512  |  512  |  1024 |  1024 |  2048 |  2048 |
> |---------------------|:------:|:--------:|:-----:|:-----:|:-----:|:-----:|:-----:|:-----:|:-----:|:-----:|:-----:|:-----:|:-----:|:-----:|:-----:|:-----:|:-----:|:-----:|
> | Pruning Method      |     C4 |     Wiki |    C4 |  Wiki |    C4 |  Wiki |    C4 |  Wiki |    C4 |  Wiki |    C4 |  Wiki |    C4 |  Wiki |    C4 |  Wiki |    C4 |  Wiki |
> | wanda               |  32.11 |    72.89 | 30.20 | 63.03 | 29.67 | 60.16 | 29.70 | 61.63 | 29.66 | 61.56 | 29.58 | 61.51 | 29.57 | 60.82 | 29.52 | 60.47 | 29.50 | 60.54 |
> | wanda + GD          |  35.72 |   100.41 | 21.01 | 40.92 | 19.51 | 36.43 | 18.99 | 34.86 | 18.70 | 33.96 | 18.50 | 33.71 | 18.36 | 33.55 | 18.23 | 33.05 | 18.16 | 32.88 |
> | sparsegpt           | 557.04 | 2,859.53 | 21.30 | 41.01 | 19.92 | 35.97 | 19.31 | 33.92 | 19.23 | 33.60 | 19.01 | 35.03 | 18.79 | 33.83 | 18.80 | 33.95 | 18.64 | 33.38 |
> | sparsegpt + GD      | 302.03 | 1,675.12 | 19.77 | 38.06 | 18.01 | 32.14 | 17.44 | 30.88 | 17.12 | 30.13 | 16.89 | 30.21 | 16.81 | 30.13 | 16.74 | 29.56 | 16.62 | 29.32 |
> | prox +GD            |  70.96 |   149.96 | 18.85 | 35.80 | 17.53 | 30.32 | 17.11 | 29.70 | 16.73 | 29.19 | 16.38 | 28.49 | 16.34 | 28.51 | 16.28 | 28.58 | 16.19 | 28.09 |
>
> ---
> ---
> ### **Performance on 70B scale:**
> We fully agree with your concern that for the Llama-3.1 70B models our method despite using more resources does not improve performance. *We thus would recommend to use wanda for those models*. While we aimed to be transparent about this, we will make it even clearer to not misguide researchers interested in this model particularly.
>
> Now for the question of *why* it does not work well on 70B models:
>
> Notice that we can qualitatively define a hierarchy between the methods by how much information of the Hessian matrix is used. Wanda uses only the diagonal entries (see also Theorem 1), SparseGPT uses already the full Hessian, and prox makes even more heavier use of the Hessian (see our toy experiment of Figure 2).
> We observe that on the smaller models, SparseGPT does quite consistently outperform Wanda --> It seems that the off-diagonal elements of the Hessian are relevant. It is those cases, where doubling-down on using Hessian information (through prox or masked gradient updates) further improves performance. Now on the other hand, on the 70B scale using plain Wanda seems close to optimal -- neither SparseGPT, prox, nor masked gradient updates improve there relevantly. This seems to indicate that the behavior on this scale is largely determined by the diagonal entries of the Hessian matrix -- in which case wanda (by our Theorem 1) is optimal.
> This is also an interesting discussion and will add it to the paper as well.

---

### Decision · Program_Chairs · 2025-05-01

**Decision:**

Reject

**Comment:**

Overall the work's motivation was well received by the reviewers, in particular with 2:4 sparsity being supported by many GPUs, and the interest in improving inference efficiency, and many found the theoretical analysis to be a significant contribution, and the appeal of the better-motivate methodology of the proposed method over Wanda/SparseGPT where the sparsity structure is simply enforced. Furthermore reviewers did comment that the paper was well-written and easy to follow in general. Reviewers were however much more borderline and mixed on the conclusiveness of the experimental validation of the method. In particular, many reviewers pointed to the significant differences in results demonstrated on smaller and larger models, a fact that the authors did concede they could not explain currently, and were less convinced that the extra computational cost of the proposed approach was worth the additional cost over Wanda or SparseGPT. Some reviewers also highlighted the lack of real-world acceleration results, although that was less of a consideration in the final decision. Overall given the post-rebuttal scores and discussion I believe this paper show great potential, but as-is remains borderline reject, and I encourage the authors to use the feedback from the reviewers to improve their work.